# Exploring *Potentilla nepalensis* Phytoconstituents: Integrated Strategies of Network Pharmacology, Molecular Docking, Dynamic Simulations, and MMGBSA Analysis for Cancer Therapeutic Targets Discovery

**DOI:** 10.3390/ph17010134

**Published:** 2024-01-19

**Authors:** Mallari Praveen, Ihsan Ullah, Ricardo Buendia, Imran Ahmad Khan, Mian Gul Sayed, Rahmul Kabir, Mashooq Ahmad Bhat, Muhammad Yaseen

**Affiliations:** 1Department of Zoology, Indira Gandhi National Tribal University, Amarkantak 484886, India; mallaripraveen950@gmail.com; 2Institute of Chemical Sciences, University of Swat, Main Campus, Charbagh 19130, Pakistan; ihsanmtk@uswat.edu.pk (I.U.); mian@uswat.edu.pk (M.G.S.); shahiswat@gmail.com (R.K.); 3Department of Chemical Biological Sciences, Universidad de las Américas Puebla, Puebla 72810, Mexico; ricardo.buendia96@gmail.com; 4Department of Chemistry, Government College University, Faisalabad 38000, Pakistan; imrankhan707470@gmail.com; 5Department of Pharmaceutical Chemistry, College of Pharmacy, King Saud University, Riyadh 11451, Saudi Arabia; mabhat@ksu.edu.sa

**Keywords:** TP53, HSPCB, Nf-kB1, *Potentilla nepalensis*, network pharmacology, computational studies, therapeutic targets

## Abstract

*Potentilla nepalensis* belongs to the Rosaceae family and has numerous therapeutic applications as potent plant-based medicine. Forty phytoconstituents (PCs) from the root and stem through n-hexane (NR and NS) and methanolic (MR and MS) extracts were identified in earlier studies. However, the PCs affecting human genes and their roles in the body have not previously been disclosed. In this study, we employed network pharmacology, molecular docking, molecular dynamics simulations (MDSs), and MMGBSA methodologies. The SMILES format of PCs from the PubChem was used as input to DIGEP-Pred, with 764 identified as the inducing genes. Their enrichment studies have shown inducing genes’ gene ontology descriptions, involved pathways, associated diseases, and drugs. PPI networks constructed in String DB and network topological analyzing parameters performed in Cytoscape v3.10 revealed three therapeutic targets: TP53 from MS-, NR-, and NS-induced genes; HSPCB and Nf-kB1 from MR-induced genes. From 40 PCs, two PCs, 1b (MR) and 2a (MS), showed better binding scores (kcal/mol) with p53 protein of −8.6 and −8.0, and three PCs, 3a, (NR) 4a, and 4c (NS), with HSP protein of −9.6, −8.7, and −8.2. MDS and MMGBSA revealed these complexes are stable without higher deviations with better free energy values. Therapeutic targets identified in this study have a prominent role in numerous cancers. Thus, further investigations such as in vivo and in vitro studies should be carried out to find the molecular functions and interlaying mechanism of the identified therapeutic targets on numerous cancer cell lines in considering the PCs of *P*. *nepalensis.*

## 1. Introduction

Traditional medicine has significantly contributed to modern pharmaceuticals by providing valuable leads for the creation of effective drugs to eradicate diseases, indicating the potential of natural products in the realm of drug development [1]. Furthermore, phytoconstituents (PCs) derived from natural sources offer the greater advantage of accessibility and fewer or no side effects when compared to synthetic drugs. The integration of natural medicinal resources into drug discovery holds promise in addressing the eradication of antibiotic resistance [2].

*Potentilla nepalensis*, also known as the Nepal cinquefoil, belongs to the Rosaceae family and is native to the Himalayan regions of Nepal and Tibet, finding its habitat in alpine and subalpine environments. Despite being admired for its captivating flowers and often cultivated as an ornamental plant, *P. nepalensis* is esteemed for its medicinal value. The plant’s therapeutic potential is closely linked to its abundance of secondary metabolite contents, including high amounts of phenols, flavonoids, and terpenoid-derived compounds, which are crucial for its therapeutic nature [3].

*P. nepalensis* has been traditionally employed for its medicinal properties in various contexts. It has found use in wound healing and addressing skin-related disease conditions, in addition to aiding digestion and promoting gastrointestinal well-being. Its potential as an anticarcinogenic agent has also been recognized [4], and its antioxidant activity contributes to safeguarding cells from oxidative stress. Moreover, the plant possesses anti-inflammatory and analgesic properties, rendering it effective in alleviating pain and mitigating inflammation [3]. Within Tibetan traditional medicine, some species of the Potentilla genus, including *P. nepalensis,* have been utilized to treat ailments such as asthma, headache, dysentery, and the common cold. Root extracts of *P. nepalensis* have exhibited promising anti-cancer [5] as well as anti-microbial activity [6].

In the present study, we examine the exploration of n-hexane and methanolic extracts obtained from both the root and stem parts of the *P. nepalensis* using the Gas Chromatography–Mass Spectrophotometry (GC-MS) method determined in previous research [7]. Our investigation aims to expose the extracts’ impact on human genes through an integrated approach of network pharmacology, molecular docking, and dynamic simulation methodologies.

This study encompasses several sequential steps. Firstly, we retrieved the pertinent information regarding the phytoconstituents (PCs) from the PubChem database. These PCs were scrutinized to identify respective inducing genes exhibiting significant pharmacological activity greater than 0.8. Subsequently, we generated protein–protein interaction networks for each extract using String DB. Our focus was then directed towards identifying the most important gene within these networks. Moreover, we performed molecular docking using Autodock Vina software. To facilitate this, we retrieved key proteins identified within the networks from the Protein Data Bank (PDB). Then, molecular dynamics simulations (MDSs) and Molecular Mechanics with Generalized Born and Surface Area solvation (MMGBSA) free energy calculations of the complexes that exhibited better binding affinity values were calculated. The detailed information on the methodology employed in this research work is presented pictorially in Figure 1.

## 2. Results

### 2.1. Retrieving the Compounds

The n-hexane and methanolic extracts obtained from the root and stem sections of *P. nepalensis*, utilizing the Gas Chromatography–Mass Spectrophotometry (GC-MS) method, were previously detailed in our study. The findings from this earlier research revealed a total of forty compounds, with ten compounds identified in each extract from both plant parts. The compounds in our study were identified with details such as their SMILES format, PubChem IDs, and 2D structures retrieved from the PubChem database. We classified these compounds into four categories based on the extraction method used for different parts of the plant. For instance, compounds labeled as **1a**–**1j** originate from the n-hexane root extract, **2a**–**2j** from the methanolic stem extract, **3a**–**3j** from the n-hexane root extract, and **4a**–**4j** from the methanolic shoot extract of *P. nepalensis*, as presented in Appendix A.

### 2.2. Inducing Genes of the PCs

A comprehensive assessment of gene induction revealed a total of 764 genes that were induced. The derived root part methanolic (MR) extract led to the induction of 149 genes. Similarly, the shoot methanolic (MS) extract resulted in the induction of 217 genes. Furthermore, the n-hexane extract obtained from the root (NR) triggered the induction of 277 genes. Lastly, the n-hexane extract from the stem (NS) extract induced 121 genes. These findings were comprehensive understandings of gene induction associated with PCs of *P. nepalensis.*

### 2.3. Enrichment Analysis

For precision and relevance, we selected the top ten descriptive terms from the comprehensive gene ontology study results. This curation focused on terms with high accuracy, as they efficiently predict gene attributes (refer to Appendix A). The descriptors were then outlined, offering insights into the genes associated with attribute description among the maximally enriched set.

The biological process (BP) indicates which gene product is involved in biological work [8]. From the extracts, a maximum number of genes’ biological processes were predominantly around two BPs: nucleic-acid-templated transcription positive and DNA-templated transcription regulation in Table 1, Appendix A. The MS extract exhibited a pronounced involvement in these two BPs.

Molecular function (MF) refers to gene product activity [8]. A converged MF emerged within the induced genes from the MR and NR extracts. Especially, protein homodimerization and protein serine/threonine activity were common denominators, effectively linked between these extracts. Expanding on this, MS and NS showed similar attributes in their induced genes. In addition, MF was enriched with MR-NR- and MS-NS-extract-induced genes. DNA binding in NR, DNA-binding transcription activator activity, phosphatase activity, and oxidoreductase activity in NS-induced genes were additional MF results in Table 1, Appendix A. This interconnectedness suggests the potential areas of research focus within the study.

Cellular components (CCs) represent the location of the gene in the cell [8]. The commonality shared between the MR- and NR-extract-induced genes is reflected in their location within the intracellular membrane-bounded organelle and nucleus. Moreover, the MR extract demonstrates an additional secretory granule lumen. Moving to the MS-extract-induced genes, these are mostly found in the nucleus and azurophil granule lumen. As for NS extract, the induced genes exhibit a more diverse localization in the intracellular organelle, endoplasmic reticulum, and secretory granule lumens in Table 1, Appendix A.

The genes targeted by the MR extract are involved in several significant pathways, such as Vegfa-Vegfr 2 signaling, leptin signaling, micro-RNAs in cardiomyocyte hypertrophy, and B-cell receptor signaling pathways. Nuclear receptors meta-pathway and Vitamin-D receptor pathways were common in the MS and NR targeted genes and additionally the Osteoblast differentiation pathway from the MS-extract-induced genes. Distinctly, the NS-extract-induced genes contribute to the drug addiction pathways and melanoma pyrimidine metabolism pathway in Table 1, Appendix A.

Upon scrutinizing the diseases associated with the genes induced by all extracts, a notable involvement in cancer-related diseases was found, especially in the context of neoplasm metastasis. This underscores the significance of these genes in the context of cancer progression and dissemination. This is effectively summarized in Table 1, Appendix A.

### 2.4. Protein–Protein Interactions

Protein-protein interaction networks were generated by interlinking the induced genes using String DB, resulting in the network containing nodes (called proteins) and edges (called interactions). The interactions have diverse sources from the experiment, text mining, gene fusion, co-expression, neighborhood, and databases. To achieve better insight into the constructed PPI networks, network topological parameters were employed, which include degree centrality, average shortest path length, clustering coefficient, closeness centrality, and betweenness centrality on each gene’s interaction proteins in the network in Table 2, Appendix A.

In evaluating the four PPI network topological parameters descriptors, degree centrality (DC) represents the number of interactions within the network. The average shortest path gauges the minimum number of edges required to connect between the nodes. The clustering coefficient (CC) measures the interconnections required to form a triangular sub-cluster. The shortest path numbers in a network can be measured between centrality (BC) values, which act as a transitional in switching the information. Closeness centrality (C. cen) signifies the closest node to the rest of the nodes in the network. Accounting for these descriptors, five actively interacting genes emerge as significant. These include HSPCB and NFKB1 from MR-extract-induced genes in Figure 2. TP53 is a central figure among MS-, NR-, and NS-extract-induced genes, illustrated in Figure 3, Figure 4 and Figure 5. These pivotal proteins that played prominent roles in the network are indicated with centrally positioned rectangular boxes, highlighted with a yellow background and blue fonts. Furthermore, Table 2 indicates the network topological parameter values of these key proteins, further reaffirming their status as potent therapeutic targets sourced from the *P. nepalensis* PCs. This evaluation paves into the critical genes and their connectivity, enhancing our understanding of the molecular level within the network.

Color was applied based on the number of interactions by each gene in the network. Rectangular shape, yellow back colored with blue font—exhibiting 15 interactions; octagonal shape, light blue back colored with green font—exhibiting 12 interactions; diamond shape, light green back colored with pink font—exhibiting 11 interactions; round rectangular shape, purple back colored with pink font—exhibiting 10 interactions; parallelogram shape, orange back colored with blue font—exhibiting 9 interactions; hexagonal shape, light green back colored with red font—exhibiting 8 interactions; ellipse circular shape, light blue back colored with red font—exhibiting less than 8 interactions.

Color is applied based on the number of interactions by each gene in the network. Rectangular shape, yellow back colored with blue font—exhibiting 25 interactions; octagonal shape, light blue back colored with green font—exhibiting 18 to 14 interactions; diamond shape, light green back colored with pink font—exhibiting 8 to 13 interactions; ellipse circular shape, light brown back colored with green font—exhibiting less than 8 interactions.

Color is applied based on the number of interactions by each gene in the network. Rectangular shape, yellow back colored with blue font—exhibiting 20 interactions; octagonal shape, brown back colored with green font—exhibiting 11 to 15 interactions; hexagonal shape, light pink back colored with pink font—exhibiting 9 to 10 interactions; round rectangular shape, orange back colored with purple font—exhibiting 5 to 8 interactions; ellipse circular shape, light green back colored with red font—exhibiting less than 5 interactions.

Color is applied based on the number of interactions by each gene in the network. Rectangular shape, yellow back colored with blue font—exhibiting 25 interactions; octagonal shape, light blue back colored with pink font—exhibiting 10 to 16 interactions; hexagonal shape, light green back colored with red font—exhibiting 7 to 9 interactions; rectangular shape, light pink back colored with blue font—exhibiting less than 7 interactions.

### 2.5. Molecular Docking

To explore the interactions between the PCs and potential therapeutic target proteins, molecular docking analyses were conducted on the methanolic and n-hexane extracts from the root and stem parts of *P. nepalensis.* The objective was to determine the binding affinity of the PCs concerning the proteins, namely p53 from the TP53 gene, heat shock protein from the HSPCB gene, and nuclear factor kappa light chain from the NFKB1 gene (see Table 2). In total, 40 PCs were subjected, resulting in docking scores that indicate the strength of binding interactions in Appendix A. Among the array of PCs, five of them—1b, 2a, 3a, 4a, and 4c—stood out for resulting in higher binding affinity with both p53 and heat shock proteins.

Figure 6 and Table 3 indicate the binding affinity and amino acid involved in interaction types. The binding affinity of 1b with the p53 protein was −8.6 kcal/mol. This interaction involved a single conventional hydrogen bond with the Aser1503 residue. Furthermore, two alkyl bonds formed, connecting the alkyl ends of 1a with the alkyl groups of BMet1584 in the p53 protein. Seventeen π-alkyl bonds emerged, linking the π-alkyl groups of ^4^ATrp1495, ATyr1502, ^2^APhe1519, ATyr1523, ^3^BTrp1495, ^3^BTyr1502, ^2^BPhe1519, and BTyr1523 with the π-orbitals of 1b. PC 2a demonstrated −8.0 kcal/mol of binding affinity with the p53 protein. AMet1584 participated in an alkyl bond formation with alkyl ends. Additionally, a set of twelve π-alkyl bonds connecting the π-alkyl groups of ^2^ATrp1495, ^2^ATyr1502, ^2^APhe1519, ^2^BTrp1495, BTyr1502, ^2^BPhe1519, and BTyr1523 with the π-orbitals of 2a.

PCs 3a, 4a, and 4c exhibited impressive docking scores of −9.6, −8.7, and −8.2 kcal/mol with HSP protein. 3a engaged in nine interactions with the HSP residues. AMet98 is involved in three different bond types, including a sulfur bond, a π-Sigma, and an alkyl bond with 3a. Three additional alkyl bonds were formed between the alkyl ends of HSP residues (^2^ALeu107, AAla111) with 3a. The π-orbitals between APhe138 with 3a are involved to form a π-π Stacked bond in enchaining their stability. Two π-alkyl bonds merged between the π-alkyl groups of APhe138 and AVal150 and the π-orbitals of 3a. For 4a, three π-alkyl bonds and a π-π stacked bond were formed with the APhe138 residue. Additionally, two π-alkyl and two π-π shaped bonds were made by 4a with ATrp162. One π-alkyl and two alkyl bonds were formed with AMet98, followed by an alkyl bond with AVal186. Another π-alkyl and one alkyl bond connected 4a and Aleu107. Lastly, 4c shows two CHB interactions with the ATrp162 residue. Six π-alkyl bonds emerged, connecting the π-alkyl groups of APhe22, APhe170, ^2^ALeu107, AMet98, and AVal150 with the π-orbitals of 4c. Furthermore, an alkyl between the alkyl groups of 4c and AIle26 formed. Two π-π stacked (^2^APhe138) and three π-π T shaped (ATyr139, ^2^ATrp162) bonds were formed by 4c. These results provide binding modes and interactions of PCs 3a, 4a, and 4a with the HSP protein, contributing potential roles in therapeutic applications.

### 2.6. Molecular Dynamics Simulations and MMGBSA

To study the dynamic nature and protein–ligand stability of the complex in the water molecular, dynamics simulations were employed. They provide insights into interaction informatic at an atomic level. In the current study, from the results of molecular docking, best pose docking score complexes were taken for MD simulations and MMGBSA calculations. A total of five complexes were employed for MDS at 300 ns; they are (i) p53 with 1b, (ii) p53 with 2a, (iii) HSPCS with 3a, (iv) HSPCB with 4a, and (v) HSPCB with 4c. These complexes were pre-processed into three stages along with energy minimization, NPT, and NVT equilibrium. Root mean square deviations (RMSDs), root mean square fluctuations (RSMFs), and MMGBSA values were analyzed by the trajectories.

The computed trajectories RMSDs of the complexes are represented in Figure 7. For p53 protein bound with 1b (p53+2b) and 2a (p53+2a) in Figure 7A, initially to the complex p53+2a, deviations were higher compared to the p53+1b. The p53+1b complex maintained constant deviation not exceeding 5Ả. Both complexes reached equilibrium at 6000 frames (~130 ns) at 2–3 Ả. Both ligands exhibit stable complexes and did not leave the binding site during the whole simulation.

For the complexes, HSPCB protein with 3a (HSPCB+3a), with 4a (HSPCB+4a), and with 4c (HSPCB+4c) are presented in Figure 7B. Their MDS analysis represents the convergence achieved at the end of 100 ns. At the beginning of the simulation, the deviations were increasing in three complexes but not exceeding greater than 3.5 Ả in HSPCB+4c. All the three complexes’ protein–ligand (s) simulations were stable between 2–3 Ả. This indicates the stability of the complexes and without leaving of ligands are exhibited during simulation from their binding regions.

RMSF by B-factor was employed to determine residue displacement in Figure 8. The RMSF of complexes, p53+1b, and p53+2a exhibited similar RMSF without major fluctuations in Figure 8A, suggesting that the binding site was not affected differently by any of both ligands. The three complexes are HSPCB+3a, HSPCB+4a, and HSPCB+4c. Their RMSF in Figure 8B shows that in the HSPCB+4c, residue 97 suffered considerable fluctuation; nonetheless, it was not involved in the binding site cavity.

Free energy calculations like MMGBSA runs performed by the cpptraj module are employed in analyzing the trajectories of the five complexes. For p53+1b and p53+2a, frame 6000 was declared as starting with five frame intervals. For HSPBC+3a, HSPBC+4a, and HSPBC+4c, frame 4000 was also started with five frame intervals. Frames taken for analysis relied on autocorrelation for every complex. Energy fluctuation calculated by MMGBSA demonstrated the difference between p53+1b and p53+2a complexes since p53+1b revealed better binding energies. Nonetheless, at the end of the simulation, these energies began to fall somehow equal to the p53+2a ligand in Figure 9A. Both ligands are considered sufficiently thermodynamically favorable to the target receptor; the *t*-tests for both complexes demonstrate a significant difference between energies in Appendix A. MMGBSA calculations demonstrated the ability to target HSPCB protein while maintaining thermodynamically favorable energies during the whole simulation. One-way ANOVA analysis demonstrates that ligand 3a is significantly different from 4a and 4c in Figure 9B; nevertheless, 4a and 4c are not different from each other in Appendix A. The summary of the MMGBSA free energies results of the complexes was presented as box-and-whisker plots in Figure 10.

## 3. Discussion

Considering the root and stem PCs by the n-hexane (NR and NS) and methanolic (MR and MS) extracts through Gas Chromatography-Mass Spectrometry (GC-MS) on *P. nepalensis*, the present study has proceeded to identify the inducing genes/protein as therapeutic targets by enrichment analysis, protein–protein interaction networks, and molecular docking followed by MDS and MMGBSA studies.

Of the total 764 inducing genes in Appendix A, prediction from the DIGEP-Pred was carried out at Pa > 0.8. Enrichment analysis was performed to find the functional annotations of the genes identified by descriptors of gene ontology, pathways, diseases, and drugs, subjecting each extract to StringDB to build PPI networks.

Through the protein–protein interaction (PPI) network, the gene-encoded proteins and their functions and interactions can be defined. In the network of PPI, proteins are represented by nodes and connecting lines referred to as edges. The current study intended to comprehend the interplay among the genes. The network topological analysis of the more interacting genes in the network is shown in Table 2. DC of HSPCB and NFKB1 is 15 of the MR-extract-induced genes, less than the TP53 from the rest of the extract-induced genes. Three proteins have average shortest path values in the range of 1.75 to 3.02, which means the connected nodes are interconnected nearer compared to the other genes. The CC of TP53 is relatively less than HSPCB and NFKB1. The C. cen values of TP53 (0.53, 0.42 and 0.57) are a little far with GSPCB and NFKB1 (0.33 and 0.34). TP53 has a greater BC value of 0.56, 0.58, and 0.59, compared to HSPCB and NFKB1 of 0.18 and 0.33. Targeting these three genes will have a greater significance in the network. Hence, three genes have been identified as potential therapeutic targets for the PCs of *P. nepalensis.*

HSPCB belongs to the heat shock protein 90 family, considered a pseudogene like a heat shock 90 kD protein 1 beta, which has a prominent role in signaling, gastric apoptosis, protein folding, and inflammation. Previous studies have reported its involvement in various cancer cell lines as potential targets such as breast cancer [9] and ovarian cancer tissues [10].

Nuclear factor kappa B subunit 1 (Nf-kB1) known as the transcription regulator activates in the nucleus by translocation into it by stimuli substances (cytokines and oxidant free radicals) followed by the transcription [11]. Problems in Nf-kB1 activation have been linked to numerous inflammatory diseases, while continuous inhibition affects immune cell development or delayed cell growth [12]. It is widely used as a therapeutic target in diabetic cardiomyopathy [13]. Thus, inhibition of the Nf-kB1 helps to reduce the anti-inflammatory activity of targeting Nf-kB1 in the associated pathways [14,15]. Designing the antagonist to the Nf-kB1 changes the central gene expression in the leukemic process [16].

The TP53 gene on transcription gives p53, which is a tumor suppressor protein that is involved in the regulation of cell division in an uncontrolled way [17]. p53 protein is widely known for its action in cancer regulation and is well renowned as a target for various cancer types because of its involvement in earlier causes of cancer [18]. At present, few drugs like piperdinone analogs, spirooxindole, nutlin, and isoquilinone are actively using inhibitors against the p53-Mdm2 complex [19]. Wide research studies are being conducted in designing the small molecules (ligands) targeting mutated p53 protein to restore tumor-suppressing activity [20].

Considering the above-mentioned research study statement and our study, purpose correlated in PCs of *P. nepalensis* found therapeutic targets can effectively exhibit their activity upon binding. Thus, finding the interaction between the PCs and identified therapeutic target proteins subjected to the docking studies resulted in PCs 1b and 2a, exhibiting better binding scores with p53 protein and PCs 3a, 4a, and 4c with HSPCB in Table 3. Most of the interactions made by the PCs with the p53 and HSPCB are hydrophobic interactions. Thus, designing these PCs with suitable chemical functional groups that could make hydrogen bonds might make the binding stronger. MDSs of these complexes, inferring RMSD in Figure 7 and RSMF in Figure 8, have better stability between them without higher deviations that are between 2 and 3 Ả. MMGBSA calculations in Figure 9 demonstrated the ability to target HSPCB protein while maintaining thermodynamically favorable energies during the whole simulation.

## 4. Materials and Methods

### 4.1. Retrieving the Compounds

PubChem IDs, 2D structures, and Simplified Molecular Input Line Entry System (SMILES) formats of the PCs found in *P. nepalensis* were retrieved from the PubChem database https://pubchem.ncbi.nlm.nih.gov/ (5 September 2023) [21]. These compounds were earlier identified from the root and stem parts using methanolic and n-hexane extracts GC-MS method [7]. PubChem is an open-source chemical compound library with a description.

### 4.2. Inducing Genes of the PCs

The SMILES format of the PCs was utilized to identify induced genes, meeting the criteria of having pharmacological activity greater than 0.8 (Pa > 0.8). This analysis was performed in the DIGEP-Pred web server http://www.way2drug.com/GE (10 September 2023) [22], which is built based on the prediction of activity spectra for substances (PASS), calculated by using leave-one-out cross-validation and depended on both mRNA data and protein data.

### 4.3. Enrichment Analysis

Enrichment analysis for the human genes induced by the PCs was conducted using Enrichr https://maayanlab.cloud/Enrichr/ (20 September 2023) by applying a false discovery rate (FDR) and a significant (*p*) value, both set to be less than 0.05 (FDR < 0.05; *p*-value < 0.05) [23]. Duplicate genes were removed, resulting in a refined gene set that was used as input to ascertain the respective gene’s biological process, molecular function, cellular components, pathways, diseases, and available drugs in the market.

### 4.4. Protein–Protein Interactions

The interactions among the predicted induced gene proteins were exploited using the String DB https://string-db.org/ (28 September 2023) platform. A high confidence threshold of 0.7 was employed to construct the networks. These networks are formed from various sources, including text mining, experimental data, databases, co-expression patterns, neighborhood analysis, gene fusion events, and co-occurrence patterns [24]. Subsequently, the resulting network was analyzed by sending it into Cytoscape V3.10.0. [25]. With the use of the Analyzer plugin, a protein–protein interaction network was established based on the gene set. These networks underwent further analysis to identify the key gene’s network topological parameters, including degree centrality, average shortest path length, clustering coefficient, closeness centrality, and betweenness centrality. These parameters offer valuable insights into the interactions and relationships among the genes within the network.

### 4.5. Molecular Docking

We employed our previously performed molecular docking methods in defining the binding affinities between the protein–ligand complexes [26]. The tertiary structures of the TP53-binding protein (PDB ID: 6MXY) https://www.rcsb.org/structure/6MXY (5 October 2023)” with K6M ligand (N-[3-(tert-butylamino)propyl]-3-(trifluoromethyl)benzamide), nuclear factor NF-kappa-BP (PDB ID: 3GUT) [27], and heat shock protein (PDB ID: 1UYM) with PU3 ligand (9-butyl-8-(3,4,5-trimethoxybenzyl)-9h-purin-6-amine) [28] were retrieved from the RCSB PDB https://www.rcsb.org/ (5 October 2023). All these structures were determined experimentally through the X-ray diffraction method with resolutions of 1.62, 3.59, and 2.45.

The PCs’ SMILES format obtained from PubChem and protein structures sourced from the PDB were not suitable for docking. To address this, the ligands were prepared by adding hydrogens at pH 7.4, generating 3D geometries, and the application of the MMFF94 forcefield using the Open Babel software [29], a tool that converts the file to various chemical formats. For the proteins, the extra bound ligands and water molecules were removed in the Drug Discovery studio. Subsequently, the protein structures were repaired by adding missing atoms, polar hydrogens, and Gasteiger charges using the Autodock software [30].

Grid parameters play a crucial role in defining the precise interacting positions and binding affinity of the ligands. For instance, the bound chemical compounds, such as K6M ligand of TP53 protein (X = −10.80, Y = 26.77, and Z = −3.52), heat shock protein with PU3 ligand (X = 3.60, Y = 11.13, and Z = 24.75), and the NFKB whole structure (X = 28.80, Y = −23.60, and Z = 58.23), were considered as grid centers with grid sizes considered as grid parameters in the present study to dock through the AutoDock Vina software [31]. This strategic utilization facilitated the accurate protein–ligand interactions and binding affinities in the context of this study.

### 4.6. Molecular Dynamics Simulations and MMGBSA

Computational molecular simulations for MD were performed on an Ubuntu 22.04 LTS workstation equipped with an Intel Core i7-13700k processor and an NVIDIA RTX 4080 graphics card. Amber 22 and AmberTools23 [32] packages were used to carry out all simulations and trajectory analyses. The ligands bound to p53 and HSPCB from docking complexes were separated, and any missing hydrogens were added using UCSF ChimeraX [33]. The ligand parameters were assigned using General Amber Force Field (GAFF) [32], and these were calculated through Antechamber under the AM1-BCC charge method. The amber force field FF19SB [34] was employed, along with the TIP3P water model. Neutralization was achieved by adding sodium and chlorine ions, and these parameters were assigned to the respective files for ligands, proteins, and solvents. To balance the system, a total of 30,000 steps of minimization were performed, followed by an increase in temperature to 300 K and 1 atm pressure equilibrium for 200 ps of simulation. A production simulation of 300 ns was then conducted using pmemd software, accelerated by CUDA. Molecular Mechanics Generalized Born Surface Area (MMGBSA) [35] analysis of trajectories was employed using module MMPBSA.py for binding energies calculation. Data visualization and figures were carried out using xmgrace 5.1.25 [36] and R v 4.2.3 [37].

## 5. Conclusions

*P. nepalensis* has gained widespread recognition for its therapeutic potential, attributed to the presence of specific compounds (PCs) identified in prior studies. In this investigation, we unveiled a total of 764 genes influenced by the plant. Through a meticulous analysis of protein–protein interaction (PPI) networks, three standout therapeutic targets emerged: TP53, derived from genes affected by multiple stimuli (MS, NR, and NS), and HSPCB and Nf-kB1, originating from genes influenced by MR. Noteworthy interactions were observed between two PCs, namely 1b (MR) and 2a (MS), with the p53 protein, displaying exceptional binding scores of −8.6 and −8.0, respectively. Additionally, three other PCs, 3a (NR), 4a, and 4c (NS), exhibited significant binding scores with the HSP protein, measuring at −9.6, −8.7, and −8.2. Molecular dynamics simulation (MDS) and Molecular Mechanics Generalized Born Surface Area (MMGBSA) analyses affirmed the stability of these complexes without substantial deviations and showcased favorable free energy values. Importantly, the identified therapeutic targets, TP53, HSPCB, and Nf-kB1, play pivotal roles across diverse cancer types. This thorough analysis underscores the potential of *P. nepalensis* compounds in modulating specific genes and proteins associated with cancer, promising avenues for further exploration in therapeutic applications.

## Figures and Tables

**Figure 1 pharmaceuticals-17-00134-f001:**
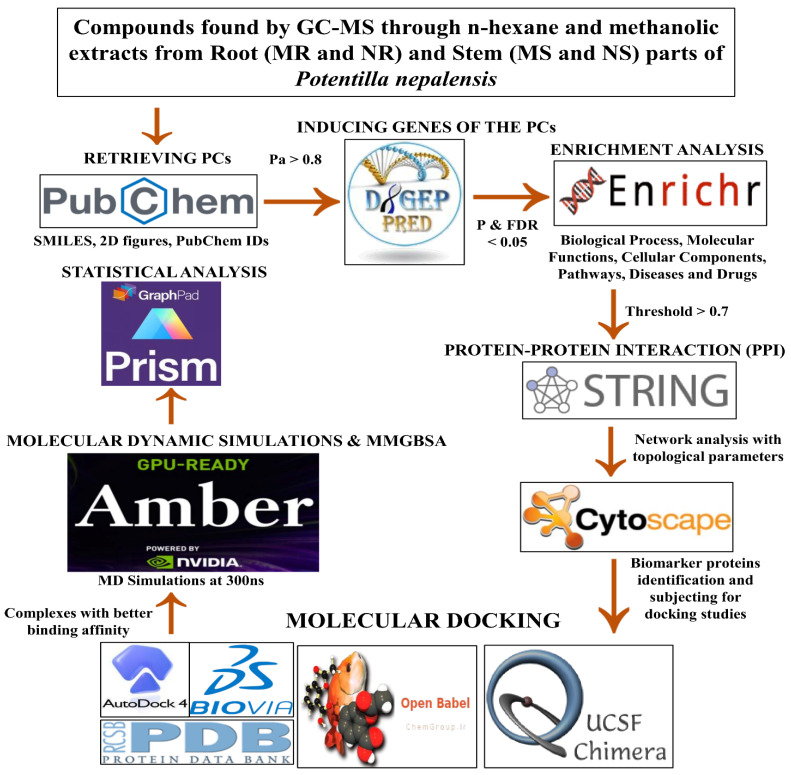
Flow chart of the strategical methodologies used in this study.

**Figure 2 pharmaceuticals-17-00134-f002:**
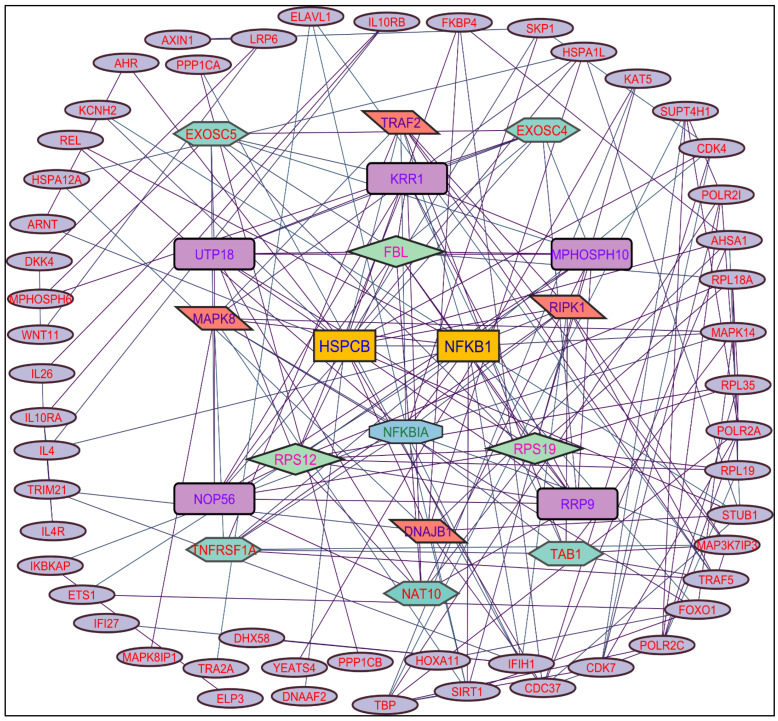
Protein–protein interaction network of the methanolic root extract-induced genes.

**Figure 3 pharmaceuticals-17-00134-f003:**
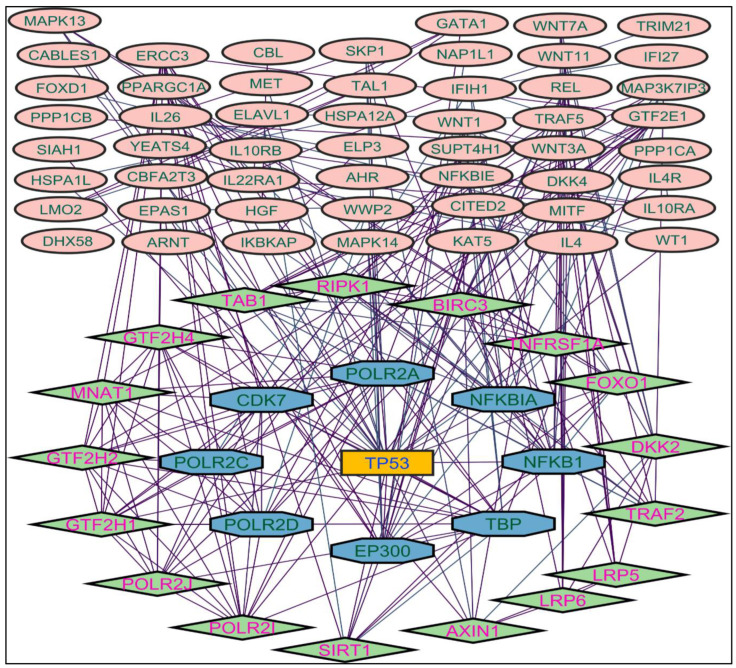
Protein–protein interaction network of the methanolic shoot extract-induced genes.

**Figure 4 pharmaceuticals-17-00134-f004:**
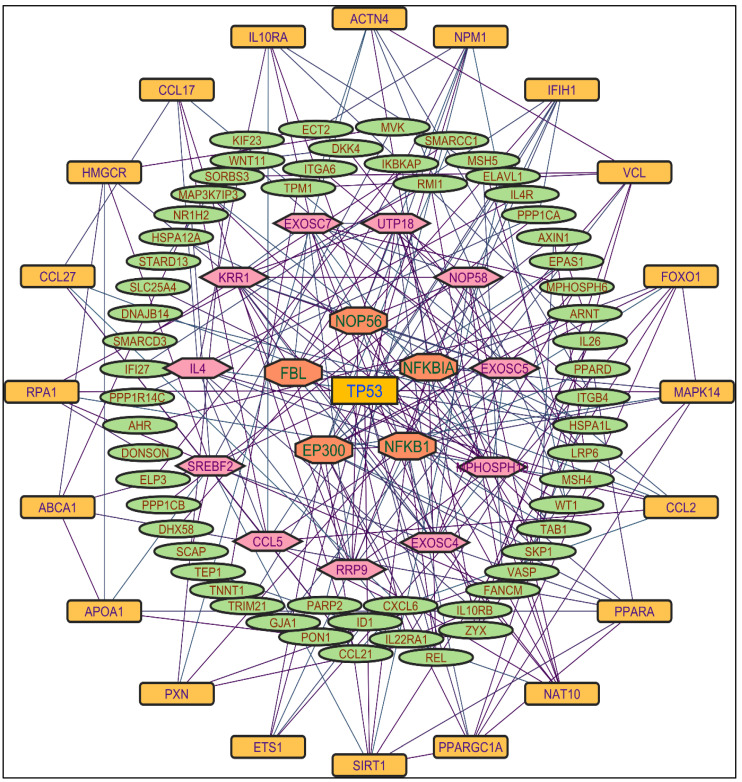
Protein–protein interaction network of the n-hexane root extract-induced genes.

**Figure 5 pharmaceuticals-17-00134-f005:**
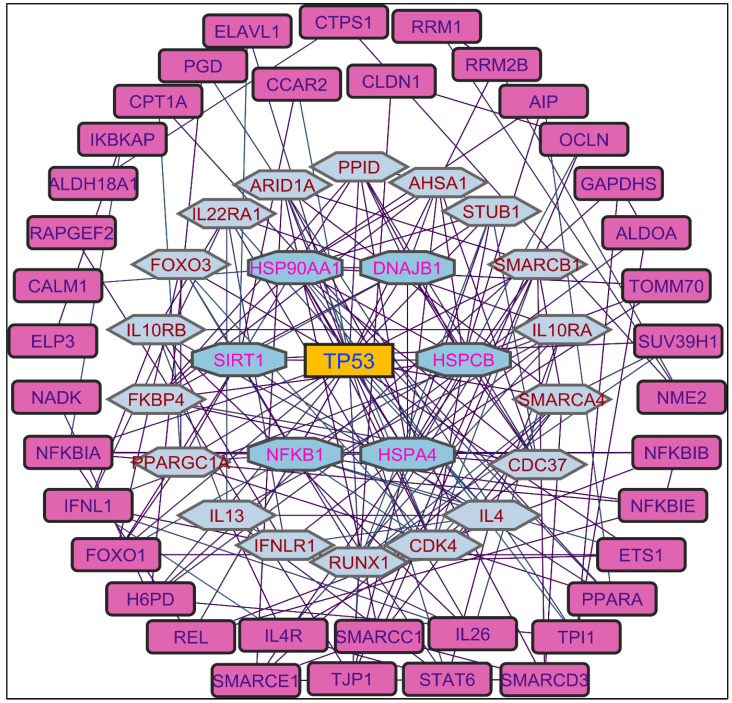
Protein–protein interaction network of the n-hexane shoot extract-induced genes.

**Figure 6 pharmaceuticals-17-00134-f006:**
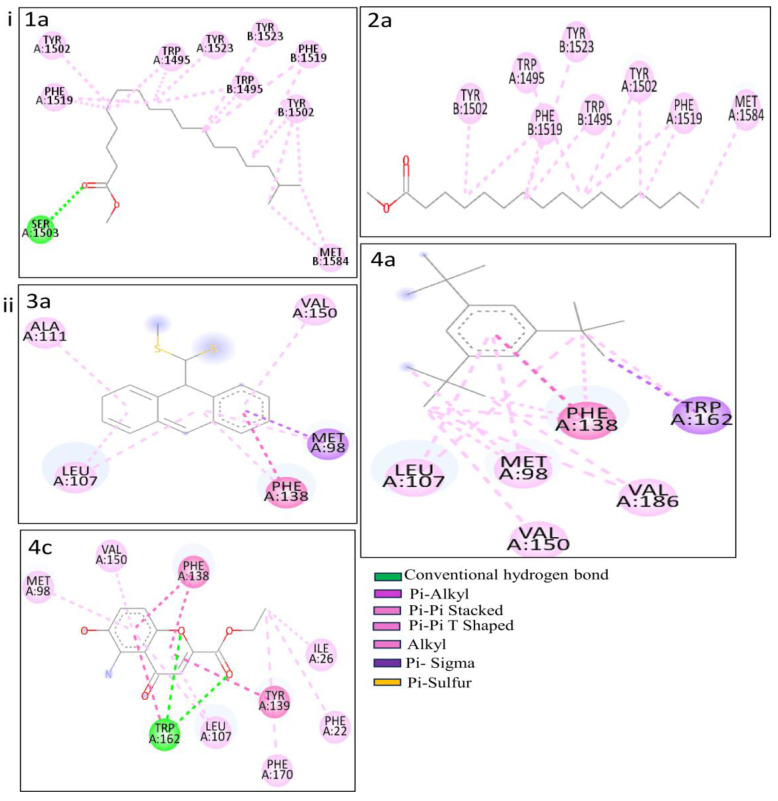
Molecular interactions of the complexes. (**i**) p53 with PCs 1a and 2a and (**ii**) HSP with PCs 3a, 4a, and 4c.

**Figure 7 pharmaceuticals-17-00134-f007:**
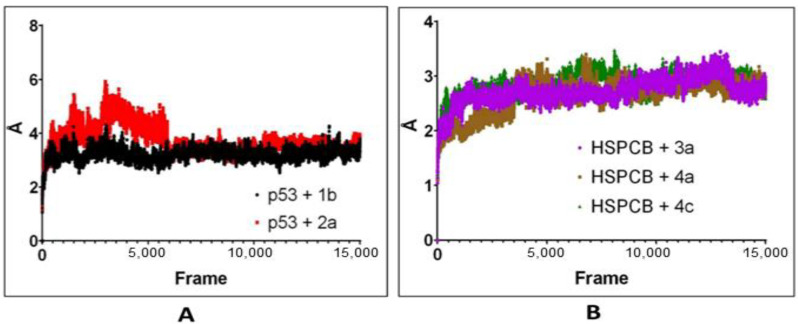
RMSD values from simulated p53 (**A**) with 1b (red) and 2a (black) complexes; HSPCB (**B**) with 3a (pink), 4a (brown), and 4c (green) complexes at 300 ns production runs divided into 15,000 frames.

**Figure 8 pharmaceuticals-17-00134-f008:**
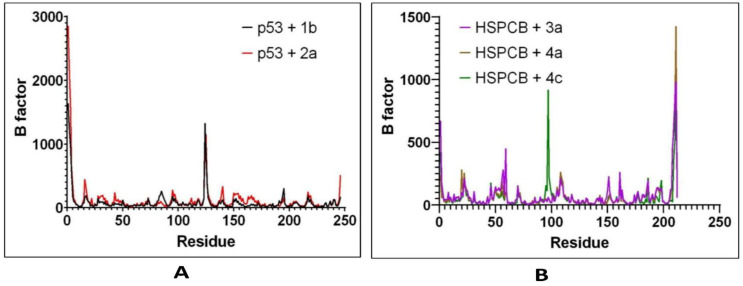
RMSF values by B-factor of every residue in p53 (**A**) with 1b (red) and 2a (black) complexes; HSPCB (**B**) with 3a (pink), 4a (brown), and 4c (green) complexes.

**Figure 9 pharmaceuticals-17-00134-f009:**
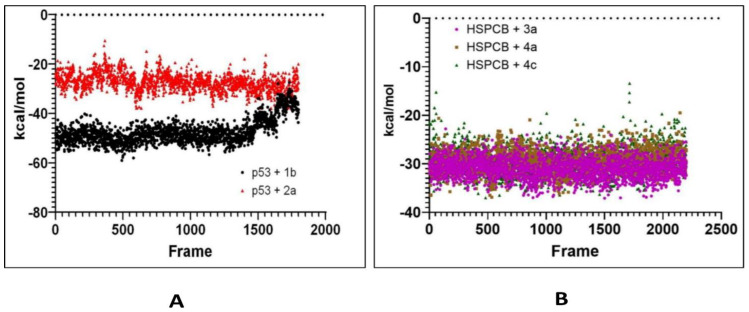
MMGBSA energy values of p53 (**A**) with 1b (red) and 2a (black) complexes; HSPCB (**B**) with 3a (pink), 4a (brown), and 4c (green) complexes.

**Figure 10 pharmaceuticals-17-00134-f010:**
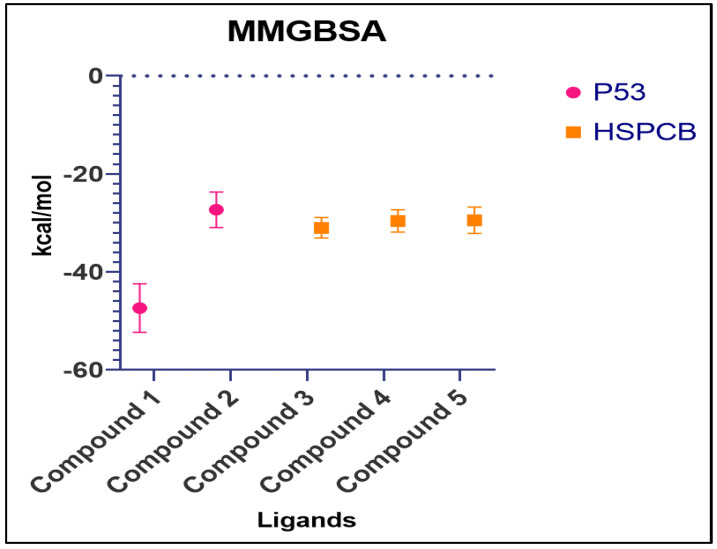
Summary of MMGBSA free energy values of both complexes, P53 (pink) and HSPCB (yellow). Compound **1**-**1b**, Compound **2**-**2a**, Compound **3**-**3a**, Compound **4**-**4a**, Compound **5**-**4c**.

**Table 1 pharmaceuticals-17-00134-t001:** Enrichment analysis of methanolic and n-hexane extracts from root and stem plant parts of *P. nepalensis*.

Descriptors	MR	MS	NR	NS
**BP**	Nucleic-acid-templated transcription positive regulation	DNA-templated transcription regulation, nucleic-acid-templated transcription positive regulation	Nucleic-acid-templated transcription positive regulation	Nucleic-acid-templated transcription positive regulation
**MF**	Protein homodimerization activity	Protein serine/threonine phosphatase activity	DNA binding, protein homodimerization activity	DNA-binding transcription activator activity, protein serine/threonine phosphatase activity, oxidoreductase activity
**CC**	Intracellular membrane-bounded organelle, nucleus, secretory granule lumen	Nucleus, azurophil granule lumen	Intracellular membrane-bounded organelle, nucleus	Intracellular organelle lumen, endoplasmic reticulum lumen, secretory granule lumen
**Pathways**	Vegfa-Vegfr2 signaling pathway, leptin signaling pathway, micro-RNAs in cardiomyocyte hypertrophy, B-cell receptor signaling pathway	Nuclear receptors meta-pathway, Osteoblast differentiation, Vitamin-D receptor pathway	Nuclear receptors meta-pathway, Vitamin-D receptor pathway	Common pathways underlying drug addiction, melanoma, pyrimidine metabolism
**Diseases**	Neoplasm metastasis, liver carcinoma, mammary neoplasms, melanoma	Neoplasm metastasis, breast carcinoma, prostate malignant neoplasm	Neoplasm metastasis, liver carcinoma	Breast carcinoma, breast neoplasm malignant, neoplasm metastasis
**Drugs**	Aprindine, Domperidone Trifluoperazine, Pitolisant, Cyproheptadine, Pimozide, Brompheniramine Buprenorphine, Lidoflazine, Chlorambucil	Mefenamic Acid, Diclofenac, Flufenamic Acid, Quercetin Mezlocillin, Hydrochlorothiazide Hydroxycarbamide Bendroflumethiazide Benzthiazide, Chlorambucil	Bezafibrate, Rosiglitazone Stearic Acid, Dodecanoic Acid, Gamolenic Acid, Aprindine, Caffeine, Eicosapentaenoic Acid, Linolenic Acid, Mefenamic Acid	Stearic Acid, Epalrestat Dodecanoic Acid, Gamolenic Acid, Vemurafenib, Bezafibrate, Gemfibrozil, Linolenic Acid, Aprindine, Eicosapentaenoic Acid

BP—biological process; MF—molecular function; CC—cellular components.

**Table 2 pharmaceuticals-17-00134-t002:** Network topological parameters of the complete protein–protein interaction network.

Induced Genes Used	Proteins	Network Topological Parameters				
		DC	Avg Short. Path Leng.	CC	C. Cen	BC
**MR**	HSPCB	15	3.02	0.18	0.33	0.18
	NFKB1	15	2.85	0.25	0.34	0.33
**MS**	TP53	25	1.87	0.15	0.53	0.56
**NR**	TP53	20	2.32	0.16	0.42	0.58
**NS**	TP53	25	1.75	0.17	0.57	0.59

DC—degree centrality; Avg Short Path Leng.—average shortest path length; CC—clustering coefficient; C. cen—closeness centrality; BC—betweenness centrality.

**Table 3 pharmaceuticals-17-00134-t003:** Binding affinity (B.A.) and interactions of the heat shock protein (HSP), nuclear factor NF-kappa-BP (NFKB1), and TP53-binding protein (p53) with the extracted PCs of *P. nepalensis*.

Complex		B.A. (kcal /mol)	Hydrogen Bonds	Hydrophobic Bonds					Other Bonds
Proteins	PCs		CHB	π-Alkyl	Alkyl	π-π Stacked	π-π T Shaped	π-Sigma	
**p53**	1b	−8.6	ASer1503	^4^ATrp1495, ATyr1502, ^2^APhe1519, ATyr1523, ^3^BTrp1495, ^3^BTyr1502, ^2^BPhe1519, BTyr1523	^2^BMet1584	-	-	-	-
	2a	−8.0	-	^2^ATrp1495, ^2^ATyr1502, ^2^APhe1519, ^2^BTrp1495, BTyr1502, ^2^BPhe1519, BTyr1523	AMet1584	-	-	-	-
**HSP**	3a	−9.6	-	APhe138, Aval150	AMet98, ^2^ALeu107, AAla111	APhe138	-	AMet98	AMet98
	4a	−8.7	-	^3^APhe138, ^2^ATrp162, AMet98, ALeu107	^2^AVal186, ^2^AMet98, AVal150, ALeu107	APhe138	-	^2^Trp162	-
	4c	−8.2	^2^ATrp162	APhe22, APhe170, ^2^ALeu107, AMet98, AVal150	AIle26	^2^APhe138	ATyr139, ^2^ATrp162	-	-

PCs—phytoconstituents; CHB—conventional hydrogen bond; other bond—sulfur bond.

## Data Availability

Data is contained within the article and Appendix A.

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
