# Peer review of "Exploring Potentilla nepalensis Phytoconstituents: Integrated Strategies of Network Pharmacology, Molecular Docking, Dynamic Simulations, and MMGBSA Analysis for Cancer Therapeutic Targets Discovery"

_pharmaceuticals, 2024, doi:10.3390/ph17010134_

Round 1
Reviewer 1 Report
Comments and Suggestions for Authors
The authors presented the paper "Exploring Potentilla nepalensis Phytoconstituents: Integrated Strategies of Network Pharmacology, Molecular Docking, Dynamics, and MMGBSA for Cancer Biomarker Discovery"
1) I don't see any author with affiliation No. 5.
2) The reference list should be improved for the Introduction section. Some old publication can be changed into a fresh one to show the progress in the area.
3) I highly recommend improving the relevance of using the extracts from these plants. Moreover, the series of reference of different authors should be presented to confirm the potential of extracts.
4) Section 2.1. Please, present compounds' names and chemical structures.
5) Section 3.1 The compounds list with percentages should be presented. Moreover, I don't see in the MDPI system any supplementary file (Supplementary Tables 1-4) with GC-MS data proved the results. Please, present supplementary data.
6) Figure 6 poor resolution and quality. Please present correct chemical structures of compounds. What is just S, N, and O with one bond? Table 3. What other bond mean?
7) Section 3.5. Please, clearly mention why you have used for docking only p53 and HSP proteins. Is this binding energy ~ -8-9 kcal/mol is enough for stable complex?
8) Figure 8. It is difficult to see that curves overlap each other.
9) Docking and MD energy calculations aree too different (-8-9 changed to ~ -30). How you explain it? Moreover, by docking the energy was -8-9 for all complound. By MD, we see -25, -45, -30. Why?
10) Conclusion section is poor. The novelty, limitations, and future outlooks of the work should be clearly mentioned in Conclusion section and Abstract.
Minor
text and formatting mistakes should be carefully proofread
Figure 7-8 bad resolution
Comments on the Quality of English LanguageModerate editing of English language required
Author Response
Reviewer 1
Comments and Suggestions for Authors
First of all, thank you so much for reviewing our manuscript and thanks again for valuable comments and suggestions.
The authors presented the paper "Exploring Potentilla nepalensis Phytoconstituents: Integrated Strategies of Network Pharmacology, Molecular Docking, Dynamics, and MMGBSA for Cancer Biomarker Discovery"
1) I don't see any author with affiliation No. 5.
Reply: thank you, affiliations have been updated.
2) The reference list should be improved for the Introduction section. Some old publication can be changed into a fresh one to show the progress in the area.
Reply: Thank you for your close observation. During preparing the manuscript and performing the study, we cited 21 articles from 2015-2023 and observed that there are only 8 research papers from 2008 to 2014 on Potentilla nepalensis. And based on your suggestions we also cited our recently published article. In Silico, In Vitro, and In Vivo Evaluation of Caffeine-Coated Nanoparticles as a Promising Therapeutic Avenue for AML through NF-Kappa B and TRAIL Pathways Modulation
- DOI: 10.3390/ph16121742
3) I highly recommend improving the relevance of using the extracts from these plants. Moreover, the series of reference of different authors should be presented to confirm the potential of extracts.
Reply: Thank you for your close observation. Since the present study is an extension to our previously performed and published work (reference no. 7), we considered only the PCs identified in that study. I consider this suggestion for the next research article.
4) Section 2.1. Please, present compounds' names and chemical structures.
Reply: Thank you for your close observation. Since, from each extract, there are 10 PCs means 40 PCs in total, used in this study. Their names, PubChem ID, SMILES formats, chemical formula, and 2D structures are mentioned in the Supplementary files (TableS1-S4)
5) Section 3.1 The compounds list with percentages should be presented. Moreover, I don't see in the MDPI system any supplementary file (Supplementary Tables 1-4) with GC-MS data proved the results. Please, present supplementary data.
Reply: Thank you for your close observation. GC-MS extract listed compounds are identified in our previous research study (reference no. 7).
6) Figure 6 poor resolution and quality. Please present correct chemical structures of compounds. What is just S, N, and O with one bond? Table 3. What other bond mean?
Reply: Thank you for your close observation. Now dpi of the Figure 6 increased and were added to the manuscript. The 2D structures of the PCs are included in Table S1-TableS4. The other bond in Table 3 description (Pg 12. Line 296) represents sulfur bond, now mentioned in the text (Pg 12. Line 311).
7) Section 3.5. Please, clearly mention why you have used for docking only p53 and HSP proteins. Is this binding energy ~ -8-9 kcal/mol is enough for stable complex?
Reply: Thank you for your close observation. The PCs of all extracts when docked with the proteins p53, Nf-kB, and HSP, these PCs have not shown good binding affinity (Table S5) and maximum interactions with the proteins. Thus, the ligands exhibited greater than -8 are considered for MDS. Regarding the complex stability after MDS, we concluded that 2 PCs, 1b and 2a with the p53 protein, and 3 PCs, 3a, 4a, and 4c, with the HSP protein, are performing stability in the complex form.
8) Figure 8. It is difficult to see that curves overlap each other.
Reply: Thank you for your close observation. Since the complexes RMSF, each receptor with its ligands in each graph. It might appear as overlap lines, but following the colour indication closely will help to read the graph.
9) Docking and MD energy calculations are too different (-8-9 changed to ~ -30). How you explain it? Moreover, by docking the energy was -8-9 for all compound. By MD, we see -25, -45, -30. Why?
Reply: Thank you for your close observation. Docking techniques are based on 2 steps. Search algorithm which objective is to generate a large cluster of “poses” or “conformations” for a ligand inside a cavity of a receptor. All poses are evaluated by a scoring function which objective is to calculate the best conformation among all of them. Most popular software relies on empirics scoring functions. That is the sum of common interactions i.e. H-bonds, aliphatic interactions, pi-pi interactions etc. That is also why some software merge steric hydrogens from molecules, since them are not capable to generate a significant interaction for empiric scoring function. The evaluation is provided in terms of free energy as the sum of all interactions found within each pose generated. Only between the ligand and the cavity.
Molecular dynamics generate trajectories of solvated state, that is, movement variations of every atom for a ligand, receptor, and a solvation box (water and ions) during a simulation time. Equivalent to scoring function, MMGBSA objective is to evaluate the complete energy contributions but for the whole solvated state. This method considers the whole system force field calculation, that is, the sum of potentials obtained by ligand contribution, receptor contribution and complex contribution with no merged atoms. The orders of magnitude for the free energy increase in contrast to docking scoring functions.
Docking technique provides a single interaction between ligand and receptor. Molecular dynamics are interactions during simulation time. Energy fluctuations are normal results. Which is expected is to maintain thermodynamical stability during long periods of simulation so the probability that ligand might take effect over receptor increases.
https://doi.org/10.1007/978-3-030-95895-4_1
The AMBER 2023 Reference Manual (https://ambermd.org/Manuals.php)
10) Conclusion section is poor. The novelty, limitations, and future outlooks of the work should be clearly mentioned in Conclusion section and Abstract.
Reply: Thank you for your close observation. Considering comments the conclusion is rewritten with the given guidelines (Pg 17. Line 448 - 464).
Minor
Text and formatting mistakes should be carefully proofread
Reply: Thank you for your close observation. As per suggestion, required edits are made now.
Figure 7-8 bad resolution
Reply: Thank you for your close observation. Now dpi of the Figures 7 and 8 increased and was added to the manuscript.

Reviewer 2 Report
Comments and Suggestions for Authors
This article is covering some aspects of Potentilla nepalensis phytoconstituents and its therapeutic application as potent plant-based medicine. Specifically, the paper is describing integrated strategies of network pharmacology, molecular docking and molecular dynamics simulations. Phytochemicals, present in n-hexane and methanolic extracts from roots and stems were analyzed by GC-MS.
The specific aims of this article and the general concept of the strategical studies are well defined and exclusively directed on identification if specific biomarkers, with potential diagnostic importance in modern medicine.
The article is concluded with a collection of 38 mostly recent references. Additionally, all 3 tables and important figures 1-10 are clear justification for the investigation of protein- protein interaction network of hexane extract (inducted 277 genes) and methanolic extract (217) induced genes. All the presented data constituted the important goals and novelty of this paper.
The following suggested changes and recommendations should be introduced before the publication of the manuscript.
- Page 2. Line 66. Replace “delve into” with “examine”
- Page 2. Line 69. Replace “unveil” with “expose”
- Page 2. Line 72. Remove “methodology”
- Page 3. Figure 1. Legend. Insert “strategical” in front of “methodologies”
- Page 7. Line 248. Replace “demarcated” with “determined”
- Page 7. Table 2. Should be moved after Figure 2. Where protein-protein interaction network particularly HSPCB and NFKB1 proteins, which indicate a potent biomarker.
- Page 11. Line 294. Insert “(see Table 2.)” in front of “In total”
- Page 12. Table 3. Should be moved to line 300, where is first discussed.
- Page 17. Line 452. Conclusion as single paragraph, regrettably is not fully informative of all strategical studies and must be significantly expanded to cover all data. Last sentence “The biomarkers identified’… should be listed in parenthesis after “identified”
- It would be extremely important to include the list of all the abbreviation used in the text as separate paragraph (after conclusions) for quick identification, description and definition.
The manuscript is of good quality and importance, and is written and edited in order to meet the standard for the articles published in Pharmaceuticals. Thus, I recommend it for publication after the correction of these suggested minor changes and recommendations.

Author Response
Reviewer 2:
Comments and Suggestions for Authors
First of all, thank you so much for reviewing our manuscript and thanks again for valuable comments and suggestions
This article is covering some aspects of Potentilla nepalensis phytoconstituents and its therapeutic application as potent plant-based medicine. Specifically, the paper is describing integrated strategies of network pharmacology, molecular docking and molecular dynamics simulations. Phytochemicals, present in n-hexane and methanolic extracts from roots and stems were analyzed by GC-MS.
The specific aims of this article and the general concept of the strategical studies are well defined and exclusively directed on identification if specific biomarkers, with potential diagnostic importance in modern medicine.
The article is concluded with a collection of 38 mostly recent references. Additionally, all 3 tables and important figures 1-10 are clear justification for the investigation of protein- protein interaction network of hexane extract (inducted 277 genes) and methanolic extract (217) induced genes. All the presented data constituted the important goals and novelty of this paper.
The following suggested changes and recommendations should be introduced before the publication of the manuscript.
- Page 2. Line 66. Replace “delve into” with “examine”
Reply: Thank you for your close observation. As per suggestion, the edit is made now
- Page 2. Line 69. Replace “unveil” with “expose”
Reply: Thank you for your close observation. As per suggestion, the edit is made now
- Page 2. Line 72. Remove “methodology”
Reply: Thank you for your close observation. As per suggestion, the edit is made now
- Page 3. Figure 1. Legend. Insert “strategical” in front of “methodologies”
Reply: Thank you for your close observation. As per suggestion, the edit is made now
- Page 7. Line 248. Replace “demarcated” with “determined”
Reply: Thank you for your close observation. As per suggestion, the edit is made now
- Page 7. Table 2. Should be moved after Figure 2. Where protein-protein interaction network particularly HSPCB and NFKB1 proteins, indicate a potent biomarker.
Reply: Thank you for your close observation. As per suggestion, the edit is made now
- Page 11. Line 294. Insert “(see Table 2.)” in front of “In total”
Reply: Thank you for your close observation. As per suggestion, the edit is made now
- Page 12. Table 3. Should be moved to line 300, where is first discussed.
Reply: Thank you for your close observation. As per suggestion, the edit is made now
- Page 17. Line 452. Conclusion as single paragraph, regrettably is not fully informative of all strategical studies and must be significantly expanded to cover all data. Last sentence “The biomarkers identified’… should be listed in parenthesis after “identified”
Reply: Thank you for your close observation. As per suggestion, the edit is made now by detailing the key findings of the current study in conclusion (Pg 17. Line 448 - 464).
- It would be extremely important to include the list of all the abbreviation used in the text as separate paragraph (after conclusions) for quick identification, description and definition.
Reply: Thank you for your close observation. As per suggestion, the edit is made now (Pg 17. Line 465 - 487).
The manuscript is of good quality and importance, and is written and edited in order to meet the standard for the articles published in Pharmaceuticals. Thus, I recommend it for publication after the correction of these suggested minor changes and recommendations.

Round 2
Reviewer 1 Report
Comments and Suggestions for Authors
Thank you for the revised version of the paper.
Comments on the Quality of English LanguageMinor editing of English language required
Author Response
Dear Reviewer,
Thank you for your valuable feedback. We have carefully implemented the suggested changes, which have been highlighted for your convenience. We believe these revisions address your concerns effectively. We hope the manuscript now meets your expectations, and we appreciate your time and insights.
Best regards,
Muhammad Yaseen